# Migration Behavior of Inclusions at the Solidification Front in Oxide Metallurgy

**DOI:** 10.3390/ma16124486

**Published:** 2023-06-20

**Authors:** Chunliang Yan, Fengming Wang, Wenling Mo, Pengcheng Xiao, Qingjun Zhang

**Affiliations:** 1College of Metallurgy and Energy, North China University of Science and Technology, Tangshan 063210, China; yanchunlinag@ncst.edu.cn (C.Y.); xiaopc@ncst.edu.cn (P.X.); 2Comprehensive Testing and Analyzing Center, North China University of Science and Technology, Tangshan 063210, China; 3College of Science, North China University of Science and Technology, Tangshan 063210, China; wfm77@ncst.edu.cn; 4College of Yisheng, North China University of Science and Technology, Tangshan 063210, China; 5Hebei Province High-Quality Steel Continuous Casting Engineering Technology Research Center, Tangshan 063210, China

**Keywords:** in situ observation, solidification front, inclusion migration behavior, microstructure

## Abstract

Distribution of inclusions plays an essential role in inducing intracrystalline ferrite, and the migration behavior of inclusions during solidification has a significant influence on their distribution. The solidification process of DH36 (ASTMA36) steel and the migration behavior of inclusions at the solidification front were observed in situ using high-temperature laser confocal microscopy. The annexation, rejection, and drift behavior of inclusions in the solid–liquid two-phase region were analyzed, providing a theoretical basis for regulating the distribution of inclusions. Analysis of inclusion trajectories showed that the velocity of inclusions decreases significantly as they near the solidification front. Further study of the force on inclusions at the solidification frontier shows three situations: attraction, repulsion, and no influence. Additionally, a pulsed magnetic field was applied during the solidification process. The original dendritic growth mode changed to that of equiaxed crystals. The compelling attraction distance for inclusion particles with a diameter of 6 μm at the solidification interface front increased from 46 μm to 89 μm, i.e., the effective length for the solidification front engulfing inclusions can be increased by controlling the flow of molten steel.

## 1. Introduction

Oxide metallurgy involves the use of micro inclusions in steel to induce intragranular acicular ferrite in crystals, forming microstructures with different orientations and cross-interlocking to improve the strength and toughness of the steel [1,2,3,4]. Presently, the excellence of acicular ferrite is unanimously recognized. Some progress has been made in the composition design of steel and the mechanism of inclusion-induced intragranular ferrite [5,6,7,8]. Although oxide metallurgy has been applied to large-wire welding technology [9], the nucleation and growth of nano- and micron-scale inclusions are not truly coupled to the macroscopic flow field.

The objective of oxide metallurgical steel is to form a uniformly dispersed system of inclusions distributed in the steel grains. The flow of molten steel and the migration behavior of inclusions during the solidification process directly determine their distribution in the solidified structure and ultimately affect the properties of the bulk material [10]. The engulfment or pushing of inclusion particles by the advancing melt–solid interface in steel is an important issue. Therefore, the migration behavior of leading-edge inclusions at the solidification interface has received widespread attention.

To enhance the inclusion of particles captured by the solidification interface into the crystal, rather than aggregating to the end of solidification and causing inclusion aggregations [11,12,13,14], many researchers have devoted themselves to studying particle behavior at the solidification frontier. Moreover, a variety of models have been developed to explain the transition between particle absorption and frontier pushing, and the dominant factors affecting particle behavior have been identified, thereby aiming to control particle behavior and distribution.

However, there are many restrictions on the application of these models owing to the complexity of the influencing system and its parameters. Research on how inclusions migrate to the solidification interface and their influencing factors during the solidification stage has been limited. Thus far, artificially controlling the behavior of particles at the forefront of the solidification interface has not been possible, significantly hindering the application of oxide metallurgy in industrial production.

Compared to traditional methods for studying the solidification process, high-temperature confocal laser scanning microscopy (HTCLSM) has a higher resolution and allows real-time in situ observation of transient phenomena at high temperatures. HTCLSM plays a vital role in research on welding, materials, and steel [15,16,17,18,19,20,21]. Dippenaar used HTCLSM to study the phase transformation processes of slag, flux, and steel during heating and cooling, correlating them with theoretical predictions [22,23,24]. He developed a concentric solidification technology based on high-temperature confocal experiments, which can be used to study the surface tension, interface morphology, and high-temperature physical properties of metallic materials [25,26]. Hiroyuki et al. discussed the mechanism of trapping and repelling inclusions using a moving solid–liquid interface during the solidification of Al-Sikilled steels [27]. However, the use of HTCLSM to study inclusion movement, behavior, and influencing factors during the solidification process of oxide metallurgical steels has not yet been reported.

Therefore, this study used HTCLSM to observe the migration behavior of inclusions in situ during the solidification process of molten steel, especially the migration at the solidification interface. From theoretical and experimental points of view, the feasibility of changing the particle distribution and the behavior of particles being engulfed or repelled at the solidification interface by changing the flow rate is discussed. It is important for improving the quality of non-modulated steel, pipeline steel, welding steel, and especially large-line energy welding steel.

## 2. Materials and Methods

### 2.1. Materials and Sample Preparation

The raw material selected was DH36 (ASTM A36) steel smelted using an oxide metallurgy process, whose chemical composition (mass fraction, %) is measured by X-ray fluorescence spectrometer (XRF) ZSX Primus II from Nippon Institute of Sciences; its composition is listed in Table 1. To obtain more explicit images, the surface of a specimen was polished to a mirror finish.

### 2.2. Experimental Apparatus and Procedure

In this study, the solidification behavior was observed in situ using a confocal laser scanning microscope equipped with an infrared furnace (VL2000DX, Lasertec, Tokyo, Japan). A halogen lamp (typically 1.5 kW) irradiated the gold-coated surface of the reaction chamber with light. The gold coating reflected energy towards the focal plane of the crucible to rapidly heat the sample, as depicted in Figure 1.

Thermocouples were installed within the Pt sample holder, which provided feedback to the halogen lamp power-supply controller. Inert crucibles are known to act partially as thermal barriers, resulting in thermocouple readings that do not reflect the actual temperature of the specimen. Thus, the experimental target temperature was calibrated using a material with a known melting point in the temperature range of interest.

Though the position temperature measured by the thermocouple of the high-temperature laser confocal microscope had a specific error with respect to the actual temperature on the surface of the steel sample, the final temperature of the experiment was determined to be 1873 K. To ensure a uniform internal temperature for the modelling, the sample was slowly heated to 1873 K at a rate of 30 K/min and held for 600 s. The cooling rate was too high, and the solidification time was too short to effectively observe the migration behavior of inclusion particles at the front of the solidification interface. Conversely, the cooling rate was too low for the induction of needle-like ferrite by oxide particles; the final cooling rate was determined to be 50 K/min. The temperature control curve of the experiment is shown in Figure 2.

The cleaned sample was placed in an alumina crucible (inner diameter of 8 mm and height of 3 mm) on a Pt holder in an infrared furnace. Before heating, the infrared furnace was alternately evacuated and backfilled with argon (Ar > 99.999%) three times to avoid the oxidation of the sample during the experiment.

A confocal laser scanning microscope was installed above the infrared image oven to observe the solidification process and inclusion behavior in the DH36 steel at high temperatures.

## 3. Results

### 3.1. Solidification Process of Molten Steel

Crystallization occurred within a specific temperature range applicable for ship plate steel with a large crystallization temperature interval. When the temperature was above the liquidus, the molten steel flowed continuously in the crucible owing to the influence of the surface tension drive; the field of view was all liquid, as shown in Figure 3a. At 1805 K, the molten steel at the observation position began to solidify. A steel cell crystal grew in a preferentially convex manner at 1805 K, forming a distinct bump, as shown in Figure 3b. New quasiparticles appeared, promoting crystal growth. The crystals that precipitated from the liquid state continued to increase in size, became thicker, and further extended in specific directions to form various dendritic morphologies, as shown in Figure 3c–e. In contrast, the amount of liquid phase in the system continued to decrease. Owing to dendrite expansion, a staggered network structure was formed, which hindered the flow of the liquid; molten steel flowed to the area with the least crystal nucleation. As the precipitated crystals in each physical region grew to form a network, the flow of liquid metal was obstructed, and the space occupied by the liquid metal continued to shrink. As the crystal grew, the liquid state completely disappeared, and the entire area eventually became solid, as shown in Figure 3f. At this time, the temperature decreased to 1773 K; the temperature difference during the solidification process was 32 K. According to the theory of component supercooling in the solidification process, the interface of the solidification front was unstable at a cooling rate of 50 K/min, perturbing the unit cell section to form a secondary dendrite arm, and finally growing as dendrites. Perturbations at the solid–liquid interface can be clearly distinguished, and coarsening of the dendrite arm can be observed. The dendrite trend is chaotic, and the spacing size is large.

During the solidification process, the faster the liquid steel flows, the easier it is for inclusions to accumulate at the end of solidification along with the remaining liquid surface flow. Inclusions do not quickly enter the austenite grains during solidification but are located at the grain boundaries. A magnetic field effectively inhibits the flow of the melt and reduces the diffusion coefficient [28,29,30,31,32]. To study the effect of the melt flow rate on the solidification process and the behavior of the inclusions, a pulsed magnetic field was applied during the solidification process [33,34]. As in the previous experiment, we observed a planar-to-dendritic transition during the solidification of the DH36 alloy melt, as shown in Figure 4a–d. However, the dendrites in the field of view precipitated almost simultaneously, the dendrite spacing increased simultaneously, and the dendrite grains became refined and orderly as shown in Figure 4e,f.

The temperature at which solidification begins was different between the two observations, though a small region was observed during the CLSM experiment. The temperature at which the crystal first appears in the field of view may not be the actual crystallization temperature. The primary phase may have been generated below the liquid surface or outside the field of view. Quasi-particles arranged on the surface likely originated from the tip of a branch of the same dendrite arm. The crystals observed in the image were most likely cross sections of the dendrites.

Comparing the in situ observations of solidified dendrites (Figure 3f and Figure 4f), the solidified microstructure of the two comparative samples is shown in the Figure 5. The untreated process produced relatively coarse dendrites. After the pulsed magnetic field treatment, the morphology changed significantly; the grains were refined, and the dendrites were neatly arranged and uniform in size.

### 3.2. Migration Behaviour of Inclusions at the Solidification Front

Using a confocal scanning laser microscope, we observed the movement of inclusions in the solid–liquid two-phase zone during an initial stage of solidification. The protrusion is a solidified phase that forms a liquid channel, in which inclusions drift along the fluid medium.

Five inclusion particles were selected for observation, inclusions a, b, c, d, and f, respectively. The positions were recorded every 0.4 s until particle motion ceased. ImageJ 1.51j8 image processing software was used to establish a coordinate axis based on the image scale and record the coordinates of the particles in each image to obtain the behavior of the particles during solidification. The trajectories of the inclusions at the solidification front are shown in Figure 6. In Figure 6d1,d2,e1,e2 refer to the two segment behavior trajectories of inclusion d and e. The inclusion particles were approximately spherical in shape; the diameters of inclusions a, b, and c are 6 µm, 5 µm, and 6 µm, respectively. The three inclusions were not immediately engulfed upon approaching the solidification interface but moved around the solidification interface. They were then engulfed either by the solidification interface or at a position away from the interface movement. Inclusion d twice approached the solidification interface frontier, but there was no solidification interface front capture. The inclusion exhibited a circular motion around the solidification interface front, and when the solid-phase region continued to expand as solidification progressed, the inclusion was finally engulfed by the solidification interface frontier. Inclusion f did not move in a circular manner around the solidification interface, maintaining a relatively long distance from the solidification interface. Instead, it moved randomly in the molten steel. As solidification progressed, new solidification interfaces continued to form and grow from the liquid surface, and inclusion e was eventually engulfed by the front of the solidification interface.

A pulsed magnetic field was applied to the molten steel to change tehe flow intensity of the liquid phase during the solidification process. The migration behavior of inclusions at the front edge of the solidification interface was observed. The migration behavior trajectories of inclusions v, w, x, and y were continuously recorded at the front edge of the solidification interface, as shown in Figure 7. Inclusions v and w exhibited irregular motion and were observed colliding and aggregating in the molten steel, migrating to form a trajectory until they were captured by the solidification interface. 

The motion trajectories of the inclusions after applying a pulsed magnetic field were significantly different from those without the applied magnetic field.

## 4. Discussions

### 4.1. Analysis of the Inclusion Movement at the Solidification Front

The particles were assumed to move at a uniform speed within a time of 0.4 s; the velocity and acceleration of the inclusions were calculated. The migration velocity and acceleration of the inclusions in the liquid phase with no applied magnetic field are shown in Figure 8.

During the solidification process, the volume fraction of the solid phase increased, the solid-phase surface gradually became uneven, and the drift rate of the inclusions decreased. The average drift rate of inclusions far from the solid–liquid interface was 200 µm/s. In contrast, the drift rate near the solid–liquid interface was relatively low, approximately 100 µm/s. The velocities of inclusions A and B decreased as the inclusions approached the solidification interface, yielding a negative acceleration. The maximum absolute value of the acceleration occurred at 50 μm from the solidification interface, where the velocity decreased to its minimum. The velocity around the solidification interface gradually increased with positive acceleration. Then, the velocity approaching the solidification interface was zero. The velocity of inclusion C increased through the solidification interface, where the acceleration was positive and maximum at 5 μm from the solidification interface. The acceleration decreased but maintained inclusion motion. Subsequently, inclusion C was attracted to another solidification interface.

Inclusion D passed through the front edge of the solidification interface many times; each time it passed through the solidification interface, the particle underwent deceleration; when approaching the solidification interface, the acceleration was negative, and the particle decelerated. When moving around the solidification interface, the acceleration was positive and increased inclusion motion. In the molten steel, inclusion F was relatively distant from the solidification interface and could not be attracted by the interface. As the solid-phase fraction continued to increase, the particles around the interface decelerated.

The small particles moved towards the solid at a velocity higher than the speed caused by the fluid flow. The effect of attraction is very clear, and the distance from which the force was effective was approximately 40 µm to 50 µm. After the inclusion approached the solid–liquid interface, it moved circularly around the precipitated solid phase and gradually moved closer to the solid phase zone. It was quickly captured, indicating that the attraction effect of the solid–liquid interface to the nearby drifting inclusions was sufficient to resist the impulse caused by inclusion drift.

The velocity and acceleration of the inclusion particles at the solid–liquid interface under the action of a pulsed magnetic field are shown in Figure 9.

The velocity and acceleration of inclusions v and w were stable during inclusion movement; the velocity suddenly increased when approaching the solidification interface until the inclusion was captured by the solidification interface. Inclusion x was eventually captured by the solidification interface when the molten steel flowed. Its velocity was nearly identical to that of the merged particles and suddenly increased when it approached the solidification interface.

This indicates that inclusion motion was affected by the force at the solidification front. When no magnetic field was applied during the solidification process, the velocity of the inclusion particles at the solidification front was approximately 200 µm/s, and the amplitude of change was large. After suppressing the melt flow by applying a magnetic field, the particle velocity was below 100 µm/s, exhibiting constant velocity. The inclusions moved with the molten steel in the initial stage of solidification. As the temperature decreased, the solidification process continued, and the inclusion particles accelerated when passing through the solidification interface. By applying a magnetic field, the flow of the melt was inhibited, and the motion, speed, and acceleration of the inclusions decreased, indicating that the flow of molten steel plays a very important role in the migration behavior of the inclusions.

### 4.2. Nature of the Long-Range Attraction of Inclusions at the Solidification Front

The force analysis on the inclusion particles is based on the laws of motion behavior. The forces on the inclusions in the fluid are those: (1) not related to the relative motion of the fluid particles, including gravity, buoyancy, etc.; (2) correlated with fluid particle relative motion or direction along relative direction of motion, including viscous resistance Stokes force and Basset force; and (3) correlated to the relative motion of the fluid particle and the direction perpendicular to the direction of motion, including the (i) Magnus force generated by the particle rotation due to the different flow velocities at different positions during the solidification of the molten steel, (ii) Magnus force from low speed to high speed when the particle is in a flow field graded by the velocity, and (iii) lateral lift Saffman force, even if there is no particle rotation. At high temperatures, the influence of the Brownian force should also be considered for inclusion particles of micron size and smaller. The forces acting on the inclusions at the solidification interface are shown in Figure 10 [35].

In this high-temperature experiment, the inclusions floated and moved on the surface regardless of gravity and buoyancy. As the inclusion size decreased, the specific surface energy increased, and the action of gravity, buoyancy, and turbulent entrainment decreased. The interaction of the particles with the solid and fluid phases plays an important role. When the inclusion particles are smaller than 10 μm, the Brownian motion is more intense, and the particles are subjected to greater Basset and Brownian forces. The velocity of the inclusions at the front of the solidification interface changed significantly, and the Saffman lift should be considered.

The Stokes viscous resistance between the inclusion particles in the molten steel and the fluid is one of the leading forces during movement. It is given by [35]:(1)$$F_{D}=C_{D}\times \frac{3\rho_{m}}{4\rho_{p}d_{p}}\left|V_{m}-V_{p}\right|\left(V_{m}-V_{p}\right),$$
where *C_D_* is the drag coefficient (a function of Reynolds number), $\rho_{m}$ and $\rho_{p}$ represent the density of molten steel and inclusions, respectively (kg/m^3^), *d_p_* represents the diameter of the inclusion particles (µm), and *V_m_* and *V_p_* represent the instantaneous velocity vectors of the molten steel fluid and inclusions, respectively.

When the inclusions move in a viscous fluid at variable speeds, the Basset force on the particle surface from the surrounding unstable fluid is given by [36]:(2)$$F_{B}=C_{B}\cdot \frac{9}{\rho_{p}d_{p}}\sqrt{\frac{\rho_{m}\cdot \mu_{eff}}{\pi }}\overset{t}{\underset{0}{\int }}\frac{\frac{d\left(V_{m}-V_{p}\right)}{d\tau }}{\sqrt{t-\tau }}\;d\tau ,$$
where *C_B_* is the Basset coefficient, *µ_eff_* is the effective dynamic viscosity of molten steel (Pa/s), and *τ* is the calculation time step (s).

The velocity of the fluid in the molten steel varied, and a pressure gradient formed between the fluids. The Saffman force applied to an inclusion under a pressure gradient is given by [37]:(3)$$F_{LS}=C_{LS}\cdot \frac{6K_{S}\mu_{eff}}{\rho_{p}\pi d_{p}}\sqrt{\frac{\rho_{m}\xi }{\mu_{eff}}}\cdot \left(V_{m}-V_{p}\right),$$
where *ξ* denotes the pressure gradient, *K_s_* is the Saffman force coefficient (1.615) [38], and *C_LS_* is the Saffman force correction coefficient.

Under high-temperature conditions, the Brownian motion of micron-sized inclusions in molten steel cannot be ignored, and the Brownian force is expressed as [39]:(4)$$F_{R}=\frac{12\delta }{\rho_{p}}\sqrt{\frac{3\mu_{eff}k_{B}T}{\pi d_{p}^{5}\Delta \tau }},$$
where *K_B_* is the Boltzmann constant (1.38 × 10^−23^ J/K), *T* is the temperature of the molten steel (K), Δτ is the time step (s), and *δ* is a random variable that follows a standard normal distribution.

From the above analysis, the drifting, engulfing, and pushing of inclusions during the solidification process of molten steel is shown to be a very complex process. During the solidification process, the volume fraction of the solid–liquid dual-phase changes, meeting the force balance of the sample where the shape of the specimen is constantly adjusted. Therefore, the molten steel in the paste area flows, promoting inclusion drift. In addition, the absorption and repulsion of relatively solid inclusions affect their movement. The accelerated movement of the inclusions indicates a strong attraction between the inclusions and the solid phase, almost perpendicular to the melt flow direction; this long-distance attraction is unrelated to surface flow [40,41].

The investigation revealed that the origin of the long-range attraction of inclusions to the solid phase could not be attributed to van der Waals forces, Rayleigh Benard flow of the liquid steel bath, or macroscopic local surface flow of liquid steel, but rather to the capillary pressure effect around the particles on the surface of the molten steel [42,43,44].

Using a schematic of the meniscus around a spherical particle at the steel–gas interface, the force between them is shown to be affected by the size of the particle and the contact angle between the particle and the liquid surface. Attraction occurs if the contact angle is greater than 90°. A single particle partially immersed in the steel melt lowers the liquid level around it. If the particle and the solid come very close together, the liquid surface between them will be further depressed or drawn downward (Figure 11).

This leads to a change in the capillary pressure between the outer region (A in Figure 11) and the inner region of the inclusion (B in Figure 11). The stress in region B is lower, and the particles are attracted to the solid–liquid interface, a mechanism known as capillary attraction.

The capillary effect is shown to be responsible for the long-range attraction of inclusions on the surface of molten steel. The contact angle between the solid particle; the steel melt, particle size, and density; and surface tension of the steel melt influences the capillary attraction force.

Until now, the capillary force between two small particles partly immersed in a liquid and its acting length have not been identified.

The total attraction force, *F*, can be given as [45]:(5)$$F=0.5\cdot g\;\left(\rho_{L}-\rho_{P}\right)\omega \Delta h^{2}.$$

Force is proportional to Δ*h*^2^, where Δ*h* is of the equilibrium difference in liquid surface height between the inside (region B) and outside (region A) of the particle. Δ*h* is given by [45]:(6)$$∆h=2\gamma cos\theta /g\left(\rho_{L}-\rho_{P}\right)d,$$
where ρ are the densities of the liquid and inclusion; *γ* is the surface tension of liquid; *d* is the separation between the solid phase and inclusion; and *θ* is the contact angle between liquid and solid inclusion. Evidently, Δ*h* is inversely proportional to *d*, and Δ*h* determines the strength of attraction.

As *d* decreases, Δ*h* increases, i.e., when the inclusions are close to the solid–liquid interface, the liquid level between them is dragged downward, and the attraction gradually increases.

Near the solid–liquid interface, the viscous resistance *F_d_* experienced by the inclusion can be expressed as [27]:(7)$$F_{d}=6\pi \eta \nu \frac{R^{2}}{d}$$

The viscous force also increases with a decrease in *d*. However, when the increase in attraction greatly exceeds the viscosity force, the inclusions are engulfed by the solid–liquid interface. There is a critical distance *d*_0_ between the inclusions and the solid–liquid interface. When the inclusions drift to a position less than *d*_0_, they are absorbed by the solid–liquid interface. In contrast, the absorption effect is not apparent.

Solid spherical inclusions were observed on the surface of the molten steel. They were driven by capillary pressure and eventually engulfed and aggregated at the solid–liquid interface. However, from the analysis of the velocity and acceleration of the inclusions, it can be seen that the forces and motions of different particles are different. Concentration gradients, temperature gradients, and surface tension at the front of the solidification interface can lead to the flow of liquids, which causes inclusions to be pushed away from the solidification interface.

The inclusions are attracted to the solidification interface when they are sufficiently close, which manifests as inclusion motion in the direction of the solid–liquid interface or flowing around the solid stage and then arranging on the solid phase. The direction of attraction to the solidification interface is determined by inclusions, and the closer to the solidification interface, the greater the attraction. The solidification interface attraction is limited, and within the effective range *d*_0_, we call it region I. In region II, the molten steel passes through the solidification interface owing to the difference in speed and the Bernoulli effect: the larger the flow velocity of the fluid, the smaller the pressure. The inclusion particles are subjected to stress along the solidification interface, resulting in a circular movement. They then leave the solidification interface front or are attracted to the solidification interface. Region III is the area sufficiently distant from the solidification interface. The inclusions are less affected by the solidification interface, instead acted upon with the same force as that for the molten steel. The force analysis of the inclusions at the front of the solidification interface and its trajectory at the solidification interface front can be inferred from the movement of the leading-edge inclusions, as shown in Figure 12a. The force analysis and actual migration behavior trajectories of the inclusions at the leading edge of the solidification interface were consistent (Figure 12b).

The effective attraction distance from the solidification interface can be measured based on the movement trajectory and motion of the inclusions at the front of the solidification interface, as listed in Table 2.

In the absence of a pulsed magnetic field, the effective attraction distance for inclusion particles with a diameter of approximately 6 μm at the front of the solidification interface was 20–60 μm. The flow velocity slowed after the application of a pulsed magnetic field. Then, the effective attraction distance for inclusions with a diameter of approximately 6 μm was 85–95 μm, nearly twice that when no magnetic field was applied. Thus, the application of a pulsed magnetic field to suppress the flow of molten steel can increase the effective attraction distance.

Enhancing the driving force and increasing the effective attraction distance can engulf inclusions at the solid–liquid interface, ensuring their uniform distribution within the solidified grain, rather than preferential accumulation at the grain boundaries. The improved mechanical properties of these materials can be fully exploited with a homogeneous distribution of the reinforcement phase. Consequently, it is necessary to control the flow and cooling rate conditions to control entrapment. Thus, the inclusion particles were engulfed by growing dendrites rather than being repelled by the interface and squeezed to the final solidification site.

## 5. Conclusions

In this study, the behavior of inclusions during the solidification of DH36 steel was observed in situ using a high-temperature laser confocal microscope. The most important conclusions of this study are as follows.

During the solidification of molten steel, heterogeneous nucleation particles were formed. As solidification progressed, the solidification interface became interconnected and staggered to create a network, and the liquid phase gradually disappeared until solidification was complete.The migration behavior of inclusions at the solidification interface can be divided into three situations: attraction, repulsion, and no effect, occupying three regions. These are region I near the interface, where the inclusions are attracted and engulfed; region II, where the inclusions are pushed by the solidification interface; and region III, where the inclusions are sufficiently far from the solidification interface, and there is no effect on the migration of inclusions. The velocity of the inclusions decreased significantly when they approached the solidification front, easing entry of the inclusions into the front.A pulsed magnetic field was applied during the solidification process to slow the melt flow speed. The solidification front interface tended to be stable, changing from the original growth mode of the dendrite to an equiaxed crystal. The compelling attraction distance of inclusions with a diameter of 6 μm at the solidification interface front increased from 46 μm to 89 μm. Therefore, the effective distance for engulfing inclusions in the solidification front can be increased by controlling the flow of the molten steel.

## Figures and Tables

**Figure 1 materials-16-04486-f001:**
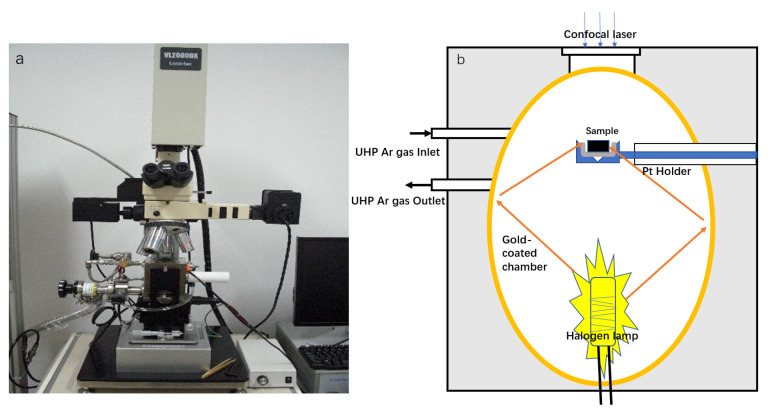
(**a**) Confocal scanning laser microscope (CLSM); (**b**) schematic of experimental apparatus.

**Figure 2 materials-16-04486-f002:**
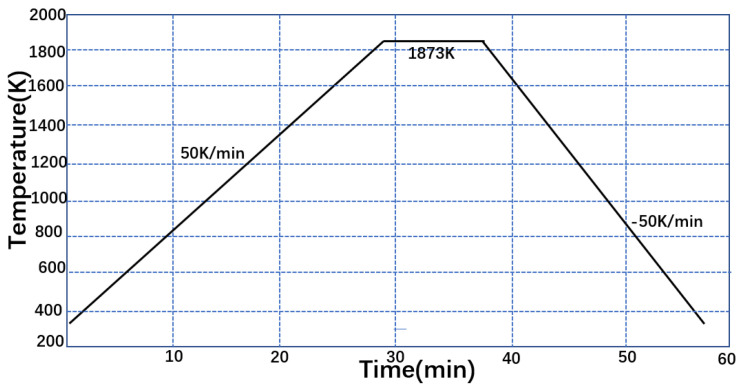
Temperature profile of the CLSM experiment.

**Figure 3 materials-16-04486-f003:**
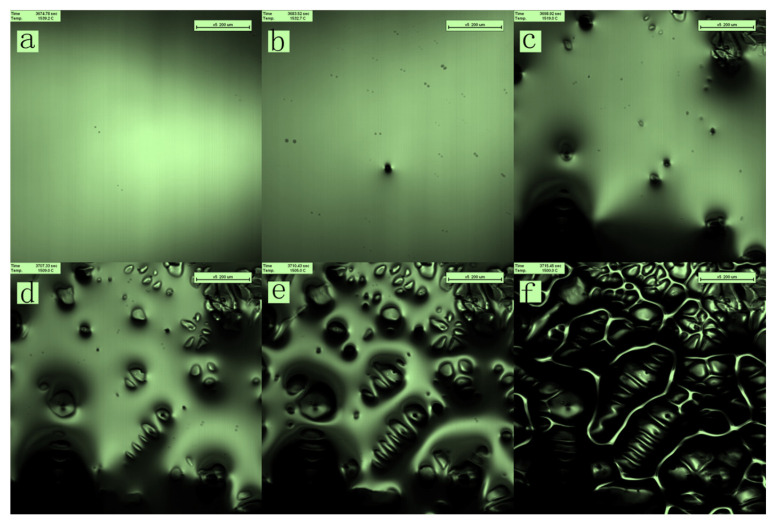
In situ observation of the solidification process.

**Figure 4 materials-16-04486-f004:**
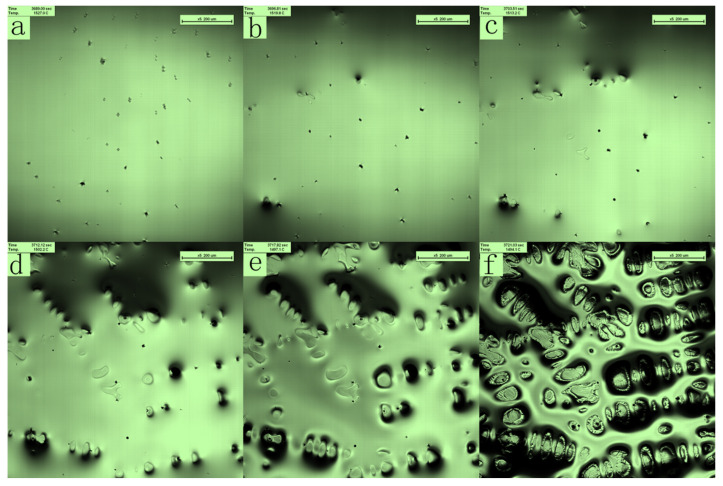
In situ observation of the solidification process when applying a magnetic field.

**Figure 5 materials-16-04486-f005:**
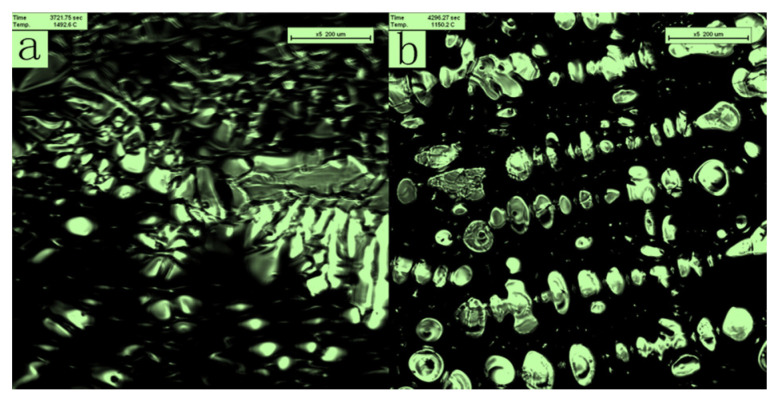
The solidified microstructure of sample (**a**) without magnetic and (**b**) with magnetic field.

**Figure 6 materials-16-04486-f006:**
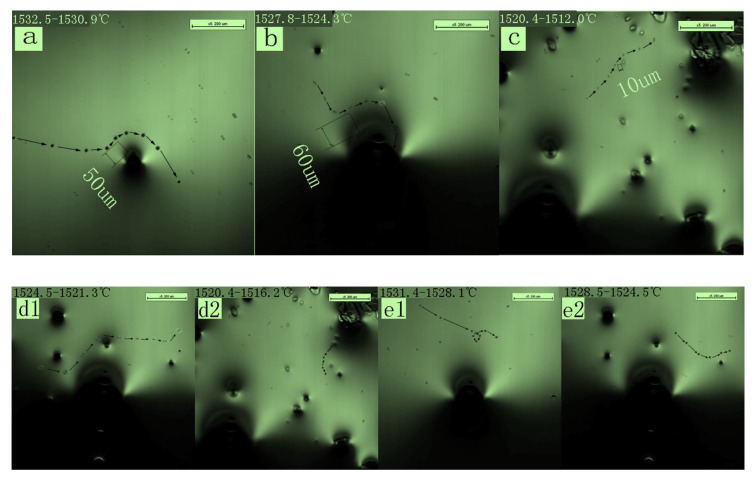
Behavioral trajectories of inclusions (**a**–**c**,**d1**,**d2**), and (**e1**,**e2**) migrating at the solidification front.

**Figure 7 materials-16-04486-f007:**
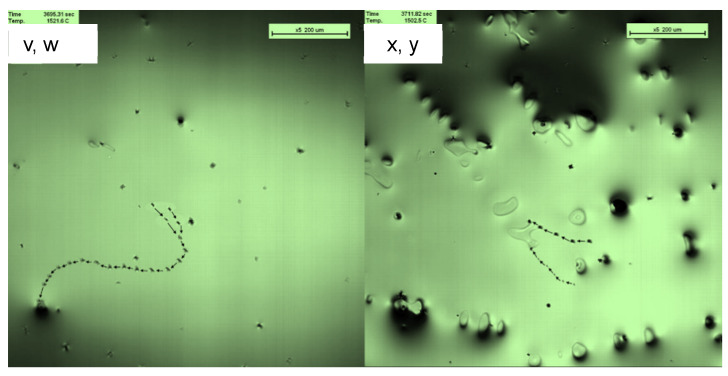
Behavioral trajectories of inclusions v, w, x, and y at the front edge of the solidification interface under pulsed magnetic field conditions.

**Figure 8 materials-16-04486-f008:**
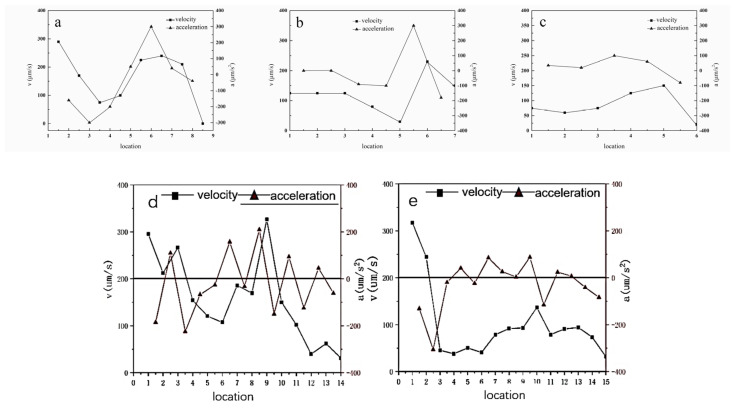
Velocity and acceleration of inclusions (**a**–**d**), and (**e**) at the solidification front.

**Figure 9 materials-16-04486-f009:**
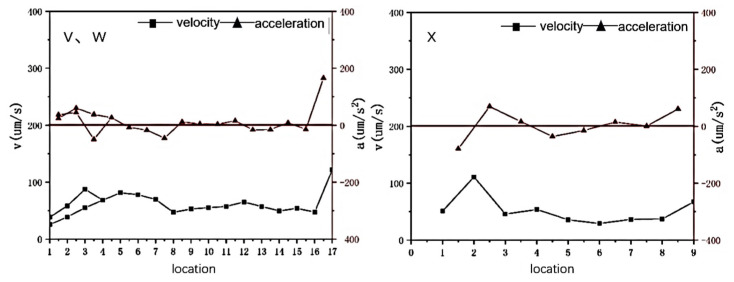
Velocity and acceleration of inclusions v, w, and x at the solidification front under pulsed magnetic field conditions.

**Figure 10 materials-16-04486-f010:**
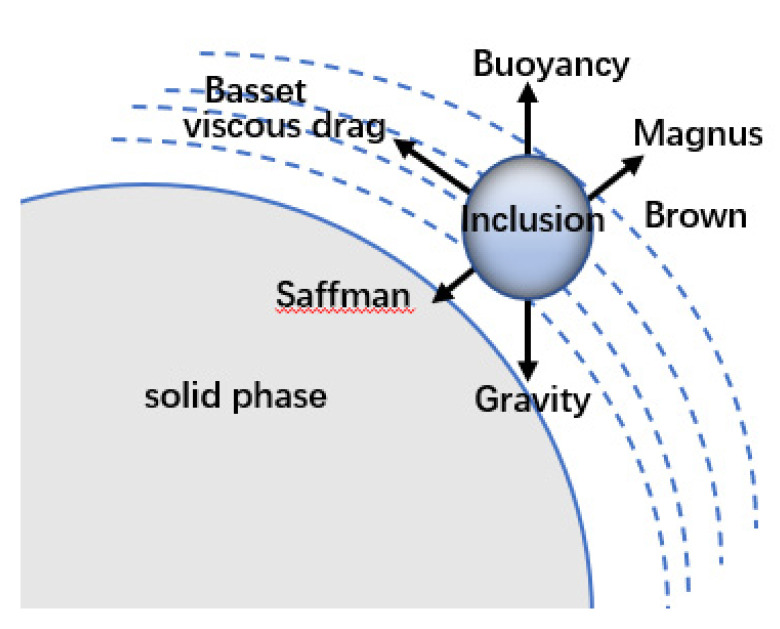
Schematic of particle–interface interaction.

**Figure 11 materials-16-04486-f011:**
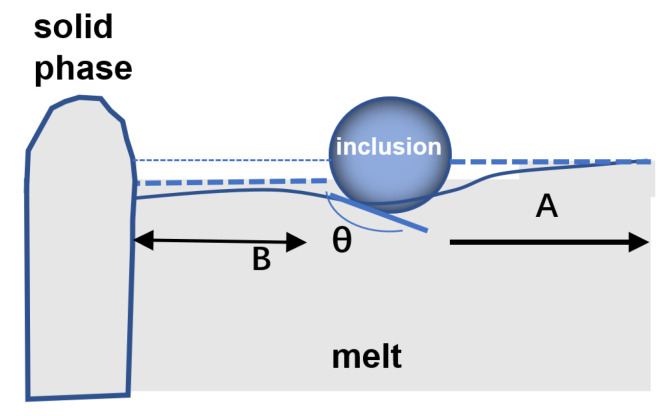
Schematic of capillary attraction between inclusions and the solid at the solid–liquid fronts.

**Figure 12 materials-16-04486-f012:**
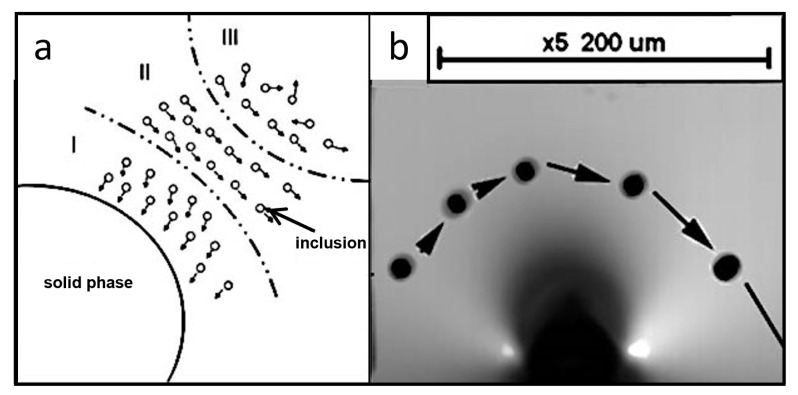
(**a**) Zoning of the frontier area of the solid–liquid interface; (**b**) movement behavior of inclusions at the front of the solidification interface.

**Table 1 materials-16-04486-t001:** Chemical composition of the experimental steel (wt%).

Element	C	Si	Mn	P	S	Al	Ti	Cr	Mg	Nb	Mo
Content (%)	0.06	0.34	1.46	0.0067	0.001	0.03	0.017	0.014	0.003	0.04	0.07

**Table 2 materials-16-04486-t002:** Effective distance from inclusion particles to solidification interface.

Particle	Diameter (µm)	Time (s)	Distance (µm)
a	6	3691.0	62
b	5	3692.5	60
c	6	3700.9	44
d	5.6	3700.4	22
e	5.8	3705.4	40
v	5.8	3696.8	93
x	6.8	3701.3	89
y	6.2	3701.8	86

## Data Availability

Not applicable.

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
