# Peer review of "Migration Behavior of Inclusions at the Solidification Front in Oxide Metallurgy"

_materials, 2023, doi:10.3390/ma16124486_

Round 1

Reviewer 1 Report

Please find the attachment reviewer's comments. 

Reviewer 2 Report

Overall, this paper contains experimental and theoretical work on the observation of the migration behavior of inclusions in situ during the solidification process of molten steel, especially the migration at the solidification interface. However, the manuscript contains some confusion, as some are stated below.:

1.     In section 1, line 32: “…the strength and toughness of the steel [1–4, 24]” Why is ref 24 highlighted? The reference should be updated

2.     In section 1, line 61: “Dippenaar used HTCLSM to study the phase transformation processes of slag, flux, and steel during heating and cooling, correlating them with theoretical predictions [22]” Ref [22] is wrongly quoted. Dippenaar is not included in Ref [22]. Likewise in line 66, “Hiroyuki et al. discussed the…” This is also different from what was presented in Ref [23]. Authors are encouraged to update all the Ref in this manuscript appropriately.

3.     Figures 3,4 and 6 should be optimized. The labeling and the numbering (a, b, c …) are difficult to read and review.

4.     In section 3.1, line 165: “After the pulsed magnetic field treatment, the morphology changed significantly; the grains were refined, the dendrites were neatly arranged and uniform in size.” Which figure shows this statement? Fig. 3h and/or 4h because the difference is not very visible.

5.      In section 3.2, line 173:” Five inclusion particles were selected for observation, inclusions A, B, C, D, and F,…” the letters (A, B, C…)  used in this statement and in Figure 5 (a, b, c,…) are different. Likewise, in line 198: “The migration behavior trajectories of inclusions V, W, X, and Y were… “ For uniformity, authors should consider using the same style and font size. The figures should be arranged neatly and of uniform scale.

6.     To bring out the beauty and clarity of the figures presented in this manuscript, is it possible to use colour imaging for figures 3, 4, 5, and 6?

7.     Figures 7 and 8 should be optimized. The axes are not clear enough and it is difficult to review. The caption should be clear enough to describe each figure. Authors should revise all the figure captions to specifically describe individual figures separately.

8.     Can authors specifically explain why presenting the velocity and acceleration of the inclusions in figures 7 and 8? Do they give different meanings and interpretations to the mobility of inclusion? If not, only one is viable.

9.     Reference is needed in Figure 9 if the figure has been reproduced by the author.

10.  A careful reading of the text should be done to suppress typo errors; can you check them, please? 

A careful reading of the text should be done to suppress typo errors; can you check them, please? 

Reviewer 3 Report

In the paper entitled „Migration behavior of inclusions at the solidification front in oxide metallurgy” Authors present the results of studies concerning mainly the example use of advanced scientific equipment i.e. high-temperature laser confocal microscopy in the studies of metallic materials in time of its solidification and crystallization. Generally, the paper is interesting and the whole presentation is clear. Moreover, the discussion of research results is on a good scientific level for the considered problem. Therefore the reviewed paper needs only minor revision. Detailed comments are given below:

#1: In the whole manuscript, the authors use the designation of studied steel as DH36. This designation results from the Chinese Standard. However, in my opinion, to increase the reach of the paper (important both for the Authors and MDPI) I suggest adding for example in brackets the equivalent designation according to ASTM or European Standards.

#2: In chapter 2.1. is a lack of data concerning spectrometer used for chemical composition measurements. Please give this data in this chapter.

#3: In table 1 please correct % on wt.%.

Author Response

Response to editor and reviewers

Manuscript Number: materials-2391462

Title: Migration Behavior of Inclusions at the Solidification Front in Oxide

Metallurgy

Dear Reviewer:

We deeply appreciate the time and effort you’ve spent in reviewing our manuscript. We have studied the editor and the reviewer’s comments carefully and have revised our manuscript. Revised portion are marked in red in our revised manuscript. The detailed corrections in the paper and the responds to your comments are listed below point by point.

Responds to the reviewer #3’s comments:

#1: In the whole manuscript, the authors use the designation of studied steel as DH36. This designation results from the Chinese Standard. However, in my opinion, to increase the reach of the paper (important both for the Authors and MDPI) I suggest adding for example in brackets the equivalent designation according to ASTM or European Standards.

Response: Thanks for your comments. We have added the equivalent name of DH36 (ASTMA36) in the article according to the comments.

#2: In chapter 2.1. is a lack of data concerning spectrometer used for chemical composition measurements. Please give this data in this chapter.

Response: Thanks for your comments. We measured the chemical composition of steel by X-ray fluorescence spectrometer(XRF) ZSX Primus II from Nippon Institute of Sciences.

#3: In table 1 please correct % on wt.%

Response: Thanks for your comments. We have modified correct % on wt.% according to the comments.

We appreciate for Editors and Reviewers’ warm work earnestly, and hope that the correction will meet with approval. If there are any problems or questions about our paper, please do not hesitate to let us know.

Once again, thank you very much for your comments and suggestions.

Reviewer 4 Report

The manuscript is very interesting.

There are only typos and not very familiar terminology used.

1. Figures 3 and 4 are missing a, b, c, d, e, f, g……., line 24 ‘forinclusion’.

2. Unfamiliar terms are used: engulfing, Al- and Si-killed steels, scaffold, circular fashion.

-

Author Response

Response to editor and reviewers

Manuscript Number: materials-2391462

Title: Migration Behavior of Inclusions at the Solidification Front in Oxide

Metallurgy

Dear Reviewer:

We deeply appreciate the time and effort you’ve spent in reviewing our manuscript. We have studied the editor and the reviewer’s comments carefully and have revised our manuscript. Revised portion are marked in red in our revised manuscript. The detailed corrections in the paper and the responds to your comments are listed below point by point.

Responds to the reviewer #4’s comments:

  1. Figures 3 and 4 are missing a, b, c, d, e, f, g……., line 24 ‘forinclusion’.

Response: Thanks for your comments. We have added the caption of Figures 3 and 4 according to the comments. ‘forinclusion’: ‘for inclusion’

  1. Unfamiliar terms are used: engulfing, Al- and Si-killed steels, scaffold, circular fashion.

Response: Thanks for your comments. We have revised these terms according to your guidance. Due to my limited abilities, I have not found a more suitable word to replace “engulfing”.

We appreciate for Editors and Reviewers’ warm work earnestly, and hope that the correction will meet with approval. If there are any problems or questions about our paper, please do not hesitate to let us know.

Once again, thank you very much for your comments and suggestions.

Yours sincerely, Qingjun Zhang

Tel.: +8603156616210

E-mail address: yanchunliang@ncst.edu.cn

Round 2

Reviewer 2 Report

The comment to manuscript materials-2391462R1 is detailed below. I did review this manuscript, and the authors attempted all the comments I raised during revision. The revision improves the quality of this manuscript. The manuscript has been well-written, and the authors have tried all the comments raised during the revision stage. Therefore, it can be published as it is in Materials.

The comment to manuscript materials-2391462R1 is detailed below. I did review this manuscript, and the authors attempted all the comments I raised during revision. The revision improves the quality of this manuscript. The manuscript has been well-written, and the authors have tried all the comments raised during the revision stage. Therefore, it can be published as it is in Materials.